# Naturally Available Flavonoid Aglycones as Potential Antiviral Drug Candidates against SARS-CoV-2

**DOI:** 10.3390/molecules26216559

**Published:** 2021-10-29

**Authors:** Ahmed A. Al-Karmalawy, Mai M. Farid, Ahmed Mostafa, Alia Y. Ragheb, Sara H. Mahmoud, Mahmoud Shehata, Noura M. Abo Shama, Mohamed GabAllah, Gomaa Mostafa-Hedeab, Mona M. Marzouk

**Affiliations:** 1Department of Pharmaceutical Medicinal Chemistry, Faculty of Pharmacy, Horus University-Egypt, New Damietta 34518, Egypt; 2Department of Phytochemistry and Plant Systematics, National Research Centre, 33 El Bohouth St., Dokki, Giza 12622, Egypt; mainscience2000@gmail.com (M.M.F.); aliayassin81@yahoo.com (A.Y.R.); monakhalil66@hotmail.com (M.M.M.); 3Center of Scientific Excellence for Influenza Virus, Environmental Research Division, National Research Centre, 33 El Bohouth St., Dokki, Giza 12622, Egypt; ahmed_elsayed@daad-alumni.de (A.M.); sarahussein9@yahoo.com (S.H.M.); shehata_mmm@hotmail.com (M.S.); noura.mahrous1995@gmail.com (N.M.A.S.); gaballah09@gmail.com (M.G.); 4Institute of Medical Virology, Justus Liebig University Giessen, 35392 Giessen, Germany; 5Pharmacology Department & Health Research Unit, Medical College, Jouf University, Skaka 11564, Saudi Arabia; gomaa@ju.edu.sa; 6Pharmacology Department, Medical College, Beni-Suef University, Beni-Suef 62521, Egypt

**Keywords:** *Anastatica hierochuntica*, *Citrus reticulate*, *Kickxia aegyptiaca*, flavonoid aglycones, molecular docking, SARS-CoV-2, *in vitro* screening

## Abstract

Flavonoids are important secondary plant metabolites that have been studied for a long time for their therapeutic potential in inflammatory diseases because of their cytokine-modulatory effects. Five flavonoid aglycones were isolated and identified from the hydrolyzed aqueous methanol extracts of *Anastatica hierochuntica* L., *Citrus reticulata* Blanco, and *Kickxia aegyptiaca* (L.) Nabelek. They were identified as taxifolin (**1**), pectolinarigenin (**2**), tangeretin (**3**), gardenin B (**4**), and hispidulin (**5**). These structures were elucidated based on chromatographic and spectral analysis. In this study, molecular docking studies were carried out for the isolated and identified compounds against SARS-CoV-2 main protease (Mpro) compared to the co-crystallized inhibitor of SARS-CoV-2 Mpro (α-ketoamide inhibitor (**KI**), IC_50_ = 66.72 µg/mL) as a reference standard. Moreover, *in vitro* screening against SARS-CoV-2 was evaluated. Compounds **2** and **3** showed the highest virus inhibition with IC_50_ 12.4 and 2.5 µg/mL, respectively. Our findings recommend further advanced *in vitro* and *in vivo* studies of the examined isolated flavonoids, especially pectolinarigenin (**2**), tangeretin (**3**), and gardenin B (**4**), either alone or in combination with each other to identify a promising lead to target SARS-CoV-2 effectively. This is the first report of the activity of these compounds against SARS-CoV-2.

## 1. Introduction

Severe Acute Respiratory Syndrome Coronavirus-2 (SARS-CoV-2) represents an evolving global threat worldwide; its infection is characterized by acute respiratory symptoms, such as fever, dry cough, and shortness of breath with an incubation period of about 5 days (average 2–14 days) [1]. By 19 October 2021, 194 vaccines are in the pre-clinical phase and 127 candidate vaccines are in clinical progress (WHO, Vaccine tracker, and landscape) [2]. Currently, different types of vaccines are approved for use in many countries, such as Sanofi–GSK, BioNTech–Pfizer, Curevac, AstraZeneca (The University of Oxford), Moderna, and Johnson & Johnson [3]. However, vaccines are a prophylactic approach and cannot be implemented for treatment, especially in pandemic situations [4]. Furthermore, the search for new therapeutic drugs from safe natural sources is crucial during pandemics [2,5,6,7,8,9,10,11,12,13]. Several naturally existing bioactive compounds were reported to behave as antiviral agents [14,15,16]. Flavonoids demonstrated antiviral and immunomodulatory activities against coronaviruses [17]. Therefore, the antiviral properties of flavonoids might also be applicable in the current COVID-19 pandemic. The antiviral activity of some flavonoids against coronaviruses (CoVs) is recognized by inhibiting 3C-like protease (3CLpro), which is capable of blocking the enzymatic activity of SARS-CoV 3CLpro [18]. 

The SARS-CoV-2 main protease (Mpro) enzyme plays an important role in the synthesis of viral functional proteins from its basic polypeptides [19,20,21]. Therefore, it seems to be responsible for both viral transcription and replication [22,23]. Based on the given facts, it is recommended to target the SARS-CoV-2 Mpro enzyme to obtain a fast and promising lead to solve the COVID-19 pandemic situation as soon as possible [24,25,26].

One of the most important methods for drug discovery processes nowadays is computational drug design [27,28]. Molecular docking studies assist scientists greatly to discover new drugs in a fast-track manner [29,30,31,32]. Moreover, molecular dynamic simulations confirm the results of molecular docking, especially in absence of *in vitro* studies [6,20]. Previous computational studies have revealed that taxifolin could be a potential inhibitor against the SARS-CoV-2 Mpro enzyme [33]. Moreover, tangeretin showed potential for the treatment and prevention of COVID-19 [34], while, hispidulin showed a better binding affinity to Mpro of SARS-CoV-2 and ACE2 receptor than hydroxychloroquine and could be used as a therapeutic candidate against COVID-19 [35]. No studies, either computational or *in vitro*, were reported for the compounds pectolinarigenin and gardenin B regarding their effects on SARS-CoV-2. Therefore, we take the responsibility for their investigations.

As an extension to our research targeting the SARS-CoV-2 Mpro enzyme [36,37,38,39], we examined the anti-SARS-CoV-2 activities of the five isolated flavonoids (**1**–**5**) and suggest their mechanism of action using molecular docking as SARS-CoV-2 Mpro inhibitors in addition to their *in vitro* evaluation.

## 2. Results and Discussion

### 2.1. Identification of the Isolated Compounds

The chemical investigation of three investigated plant extracts led to the isolation of five major flavonoid aglycones (**1**–**5**). Taxifolin (**1**) and pectolinarigenin (**2**) were obtained from *A. hierochuntica* and *K. aegyptiaca,* respectively, whereas the citrus peel extract afforded three methoxylated flavonoid aglycones—tangeretin (**3**), gardenin B (**4**), and hispidulin (**5**). Their chemical structures are shown in Figure 1.

### 2.2. Docking Studies

The study of the binding mode of the co-crystallized α-ketoamide inhibitor (**KI**) of the isolated dimer form of the SARS-CoV-2 Mpro showed an asymmetric binding. Moreover, the molecular docking of the α-ketoamide inhibitor (**KI**) was conducted in addition to the isolated and identified flavonoids, namely taxifolin (**1**), pectolinarigenin (**2**), tangeretin (**3**), gardenin B (**4**), and hispidulin (**5**) against SARS-CoV-2 Mpro. The binding scores for the docked compounds were found to be in the following order: redocked **KI** ˃ tangeretin (**3**) ˃ taxifolin (**1**) ˃ gardenin B (**4**) ˃ hispidulin (**5**) ˃ pectolinarigenin (**2**). Their binding scores were near to each other (from −6.61 to −5.74 kcal/mol) compared to that of the docked co-crystallized α-ketoamide inhibitor (−8.17 kcal/mol), with promising binding interactions with the pocket amino acids (Table 1).

Regarding the docking results depicted in Table 1, it is worth mentioning that tangeretin (**3**) showed the best binding score among all isolates (−6.61 kcal/mol) compared to the docked co-crystallized native Mpro inhibitor (**KI**, −8.17 kcal/mol). Tangeretin (**3**) was stabilized inside the Mpro pocket of SARS-CoV-2 through the formation of 2 pi-H bonds with Glu166 amino acid at 4.09 and 4.19 Å. Furthermore, the docked **KI** formed 3 H-bonds with Glu166 amino acid at 2.89, 3.10, and 3.42 Å. It also formed 1 pi-H bond with Gly143 amino acid at 3.70 Å (Table 1 and Table 2). 

It is evident that the Glu166 amino acid seems to be very crucial for SARS-CoV-2 Mpro pocket binding and inhibition.

From Table 1 and Table 2 it can be observed that the docking results of the isolated and identified five flavonoids from the aerial parts of *A. hierochuntica* and *K. aegyptiaca* and the citrus peel of *C. reticulata* fruits, namely taxifolin (**1**), pectolinarigenin (**2**), tangeretin (**3**), gardenin B (**4**), and hispidulin (**5**), examined against SARS-CoV-2 Mpro and compared to the docked **KI**, give us a clear promising idea towards their binding affinities, which indicates, subsequently, their expected intrinsic activities as well their importance to combat the SARS-CoV-2 pandemic.

### 2.3. In Vitro Validation

Based on the in silico studies, pectolinarigenin, tangeretin, and gardenin B showed the best evidence of the studied drugs to be selected for further *in vitro* validation against SARS-CoV-2. Hence, the *in vitro* study was conducted on the five compounds and the results were effective with pectolinarigenin, tangeretin, and gardenin B. To identify the proper concentrations to define the antiviral activity of pectolinarigenin, tangeretin, and gardenin B, the half-maximal cytotoxic concentration “CC_50_” was calculated by a crystal violet assay (Figure 2). All compounds showed a wide range of safety within the tested concentrations (10 ng/mL–100 mg/mL). 

The antiviral screening revealed that pectolinarigenin (**2**) and tangeretin (**3**) exhibited a promising cytotoxic inhibitory activity against NRC-03-nhCoV with IC_50_ = 12.4 and 2.5 µg/mL, respectively (Figure 2b,c). Both natural compounds exerted their anti-SARS-CoV-2 activities with high selectivity indices (CC_50_/IC_50_ > 1000). In previous reports that mentioned the biological activities of pectolinarigenin and gardenin B; pectolinarigenin showed potent inhibitory activities on melanogenesis [40] and exhibited powerful *in vitro* anti-diabetic, hepatoprotective, and anticancer activities [41,42,43]. On the same line, gardenin B, which is a methoxylated flavonoid derived from a tangeretin, showed slight anti-SARS-CoV-2 activity (IC_50_ = 128 µg/mL). Interestingly, gardenin B, motioned previously for its induction of cell death in human leukemia cells, involves multiple caspases [44] and also shows *in vitro* antiviral activity against the *Encepehalomyocarditis* virus (EMV) [45]. 

## 3. Material and Methods

### 3.1. Plant Material

Three plant species were collected and identified as belonging to three different families: *A. hierochuntica* L. (Brassicaceae), *C. reticulata* Blanco (Rutaceae), and *K. aegyptiaca* (Plantaginaceae). The aerial parts of the first and last species were collected from the northern coast of El Dabaa road, in March 2019, while the fresh matured fruits of *C. reticulata* were obtained from the traditional market, Giza, Egypt.

### 3.2. Extraction, Isolation and Structure Elucidation

The aerial parts of *A. hierochuntica* and *K. aegyptiaca* as well as the peel of *C. reticulata* fruits were air-dried and ground. Each obtained powder was extracted with MeOH:H_2_O (7:3) 3 times at room temperature. All extract were evaporated under reduced pressure and temperature to obtain residues. Each residue was subjected to an acid hydrolysis process (2N HCl, 100 °C, 2 h) [46]. The acidic solutions were extracted with ethyl acetate several times, affording aglycones extracts upon evaporation. Each extract was subjected to a Sephadex LH-20 column; using MeOH:H_2_O (1:1) afforded fractions. Each fraction was subjected to PPC using BAW and 50% AcOH several times to isolate the flavonoid aglycones. All compounds were finally purified with a Sephadex LH-20 column, using 100% MeOH as eluent to reach pure aglycones. Compound (**1**) was obtained from *A. hierochuntica,* compound (**2**) from *K. aegyptiaca*, while compounds (**3**–**5**) were obtained from *C. reticulata* (tangerine). The structures of the isolated flavonoids were elucidated by extensive chromatographic, chemical, and spectroscopic methods (HRESI–MS, UV, and NMR) as well as Co-PC with reference samples. Their spectroscopic data were compared with previously reported values [38,39,40,41]. HRESI–MS and NMR chromatograms are provided as Appendix A (Appendix A).

#### 3.2.1. Taxifolin (Dihydroquercetin) (**1**)

^1^H-NMR (DMSO-*d_6_*, 500 MHz): δ 11.87 (1H, br s, 5-OH), 6.83 (2H, m, *J* = 2.0 Hz, H-2′, H-6′), 6.69 (1H, *J* = 8.0 Hz, H-5′), 5.87 (1H, d, *J* = 2.0 Hz, H-8), 5.82 (1H, d, *J* = 2.0 Hz, H-6), 5.72 (1H, d, *J* = 6.5 Hz, H-2), 4.95 (1H, dd, *J* = 6.5 Hz, H-3_ax_), 4.45 (1H, dd, *J* = 17.0, 5.0 Hz, H-3_eq_). Positive HRMS: 305.0723 (C_15_H_13_O_7_^+^) [47].

#### 3.2.2. Pectolinarigenin (Scutellarein 4′,6-Dimethyl Ether) (**2**)

^1^H-NMR (DMSO-*d_6_*, 500 MHz): δ 13.01(1H, s, 5-OH), 10.71 (1H, s, 7-OH), 8.01 (2H, d, *J* = 8.5 Hz, H-2′, H-6′), 7.09 (2H, d, *J* = 8.5 Hz, H-3′, H-5′), 6.85 (1H, s, H-8), 6.59 (1H, s, H-3), 3.83 (3H, s, 4′-OCH_3_), 3.71 (3H, s, 6-OCH_3_). Negative HRMS: 313.0719 (C_17_H_13_O_6_^−^) [48]. 

#### 3.2.3. Tangeretin (4′,5,6,7,8-Pentamethoxyflavone) (**3**)

^1^H-NMR (DMSO-*d_6_*, 500 MHz): δ 7.98 (2H, d, *J* = 8.5 Hz, H-2′, H-6′), 7.12 (2H, d, *J* = 8.5 Hz, H-3′, H-5′), 6.73 (1H, s, H-3), 3.99 (3H, s, 5-OCH_3_), 3.94 (3H, s, 7-OCH_3_), 3.84 (3H, s, 4′-OCH_3_), 3.8 (3H, s, 8-OCH_3_), 3.74 (3H, s, 6-OCH_3_). Positive HRMS: 373.1285 (C_20_H_21_O_7_^+^) [49]. 

#### 3.2.4. Gardenin B = Demethyltangeretin (5-Hydroxy 6,7,8,4′-Tetra Methoxy Flavone) (**4**)

^1^H-NMR (DMSO-*d_6_*, 500 MHz): δ 12.51(1H, s, 5-OH), 8.01 (2H, d, *J* = 8.5 Hz, H-2′, H-6′), 7.13 (2H, d, *J* = 8.5 Hz, H-3′, H-5′), 6.78 (1H, s, H-3), 3.84 (3H, s, 7-OCH_3_), 3.83 (3H, s, 4′-OCH_3_), 3.75 (3H, s, 8-OCH_3_), 3.74 (3H, s, 6-OCH_3_). Positive HRMS: 359.1135 (C_19_H_19_O_7_^+^) [49].

#### 3.2.5. Hispidulin (**5**)

^1^H-NMR (DMSO-*d_6_*, 500 MHz): δ 13.05(1H, s, 5-OH), 10.68 (1H, s, 7-OH), 10.33 (1H, s, 4′-OH), 7.91 (2H, d, *J* = 8.5 Hz, H-2′, H-6′), 6.89 (2H, d, *J* = 8.5 Hz, H-3′, H-5′), 6.76 (1H, s, H-8), 6.56 (1H, s, H-3), 3.71 (3H, s, 6-OCH_3_). Negative HRMS: 299.0905 (C_16_H_11_O_6_^−^) [50].

### 3.3. Molecular Docking Study

The molecular docking study was performed using the MOE 2019.012 suite [51,52] for the isolated and identified five flavonoids from *A. hierochuntica*, *K. aegyptiaca*, and citrus peels, namely taxifolin (**1**), pectolinarigenin (**2**), tangeretin (**3**), gardenin B (**4**), and hispidulin (**5**), to propose their mechanism of action as SARS-CoV-2 Mpro inhibitors based on their binding scores and interactions. 

Moreover, they were compared to the co-crystallized inhibitor of SARS-CoV-2 Mpro (**KI**) as a reference standard.

#### 3.3.1. Preparation of the Isolated and Identified Five Flavonoids (**1**–**5**)

The 2D chemical structures of the isolated five flavonoids—taxifolin (**1**), pectolinarigenin (**2**), tangeretin (**3**), gardenin B (**4**), and hispidulin (**5**)—were sketched using ChemDraw Professional. Each chemical structure was introduced separately into the MOE window, converted to the 3D orientation, adjusted for partial charges, and energy minimized to be prepared for docking according to the default preparation steps described earlier [53,54,55,56]. After saving each prepared compound separately using the (.moe) extension, the co-crystallized native inhibitor of SARS-CoV-2 Mpro (**KI**) was extracted and saved in a separate MOE file as well. Furthermore, all of the aforementioned prepared compounds (**1**–**5**) were imported in the same database file and saved as (.mdb) extension to be uploaded during the docking step.

#### 3.3.2. Target Mpro of SARS-CoV-2 Preparation

The target Mpro enzyme (as a dimer) of SARS-CoV-2 was extracted from the Protein Data Bank (PDB code: 6Y2G) [57]. Moreover, it was subjected to the detailed preparation steps described before [58,59,60,61] to be ready for the docking process.

#### 3.3.3. Docking of the Database Compounds (**1**–**5**) to the Dimer Mpro of SARS-CoV-2

The previously discussed database, containing the **KI** in addition to the five isolated and identified flavonoids (**1**–**5**), was uploaded in place of the ligand during a general docking process. The binding site of the co-crystallized α-ketoamide inhibitor was identified as the docking site. Moreover, the program specifications were adjusted as follows: triangle matcher for the placement methodology, London dG for the first scoring methodology, GBVI/WSA dG for the final scoring methodology to select the best 10 poses from 30 different poses for each docked compound, and rigid receptor for the refinement methodology [61,62,63,64]. Finally, the best pose for each tested compound, based on the score and RMSD values, was selected for further studies. 

Furthermore, a MOE program validation process was carried out before applying the previously described docking process by redocking the co-crystallized **KI** alone at its binding site of Mpro. The obtained low RMSD values (<2) between the native co-crystallized and the redocked α-ketoamide inhibitor confirmed the valid performance [65,66,67].

### 3.4. In Vitro Anti-SARS-CoV-2 Activity

#### 3.4.1. Cytotoxicity (CC_50_) Determination

To assess the half-maximal cytotoxic concentration (CC_50_), stock solutions of the compounds were prepared in 10% DMSO in ddH_2_O and diluted further to the working solutions with DMEM. The cytotoxic activity of the extracts was tested in VERO-E6 cells by using a crystal violet assay, as previously described [68] with minor modifications. Briefly, the cells were seeded in 96 well-plates (100 µL/well at a density of 3 × 10^5^ cells/mL) and incubated for 24 h at 37 °C in 5% CO_2_. Control cells were treated with 1% DMSO in DMEM (the concentration of DMSO in the highest concentration of the tested samples). After 24 h, the cells were treated with various concentrations of the compounds in triplicates. After 72 h, the supernatant was discarded, and the cell monolayers were fixed with 10% formaldehyde for 1 h at room temperature (RT). The fixed monolayers were, then, dried and stained with 50 µL of 0.1% crystal violet for 20 min on a bench rocker at RT. The monolayers were, then, washed and dried, and the crystal violet dye in each well was dissolved with 200 µL methanol for 20 min on a bench rocker at RT. The absorbance of the crystal violet solutions was measured at λmax 570 nm as a reference wavelength using a multi-well plate reader. The cytotoxicity of the various concentrations, compared to the untreated cells and the blank background, was determined using nonlinear regression analysis by plotting the log inhibitor versus the normalized response.

#### 3.4.2. Inhibitory Concentration 50 (IC_50_) Determination 

The IC_50_ values for the compounds were determined as previously described [69], with minor modifications. Briefly, in 96 well tissue culture plates, 2.4 × 10^4^ Vero-E6 cells were distributed in each well and incubated overnight in a humidified 37 °C incubator under 5% CO_2_ conditions. The cell monolayers were then washed once with 1× PBS. An aliquot of the SARS-CoV-2 “NRC-03-nhCoV” virus [70] containing 100 TCID_50_ was incubated with serially diluted concentrations of the tested compound and kept at 37 °C for 1 h. The Vero-E6 cells were treated with a virus/compound mix and co-incubated at 37 °C in a total volume of 200 µL per well. Untreated cells infected with the virus represented virus control; however, cells that were not treated and not infected were cell control. Following incubation at 37 °C in a 5% CO_2_ incubator for 72 h, the cells were fixed with 100 μL of 10% paraformaldehyde for 20 min and stained with 0.5% crystal violet in distilled water for 15 min at RT. The crystal violet dye was then dissolved using 100 μL absolute methanol per well and the optical density of the color was measured at 570 nm using an Anthos Zenyth 200 rt plate reader (Anthos Labtec Instruments, Heerhugowaard, Netherlands). The IC_50_ is the concentration of the compound required to reduce the virus-induced cytopathic effect (CPE) by 50%, relative to the virus control.

### 3.5. Statistical Analyses

All experiments were performed in three biological repeats. Statistical tests and graphical data presentation were carried out using GraphPad Prism 5.01 software. Data are presented as the average of the means. The IC_50_ and CC_50_ curves represent the nonlinear fit of “normalize” of “transform” of the obtained data; their values were calculated using GraphPad prism as “best fit value”.

## 4. Conclusions

Five compounds were isolated and identified from *A. hierochuntica*, *K. aegyptiaca*, and the citrus peels of *C. reticulata*, namely, taxifolin (**1**), pectolinarigenin (**2**), tangeretin (**3**), gardenin B (**4**), and hispidulin (**5**), and examined against SARS-CoV-2 Mpro using *in vitro* and molecular docking studies. Their IC_50_ and binding score values indicate that the examined flavonoids, especially pectolinarigenin (**2**), tangeretin (**3**), and gardenin B (**4**), could be very promising for performing more advanced preclinical and clinical tests, either alone or in combination with each other, for COVID-19 management. 

## Figures and Tables

**Figure 1 molecules-26-06559-f001:**
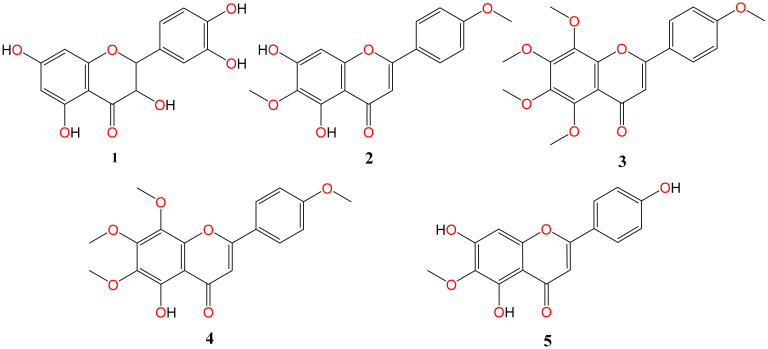
The chemical structures of the isolated flavonoid compounds.

**Figure 2 molecules-26-06559-f002:**
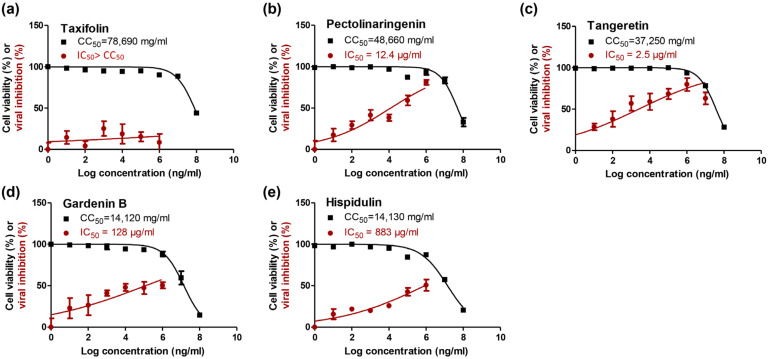
Dose-response and inhibition curves for the five isolated compounds (taxifolin (**a**), pectolinarigenin (**b**), tangeretin (**c**), gardenin B (**d**), and hispidulin (**e**)) showing the half-maximal cytotoxic concentration (CC_50_) in Vero E6 cells and inhibitory concentration 50% (IC_50_) against NRC-03-nhCoV which were calculated using the nonlinear regression analysis of the GraphPad Prism.

**Table 1 molecules-26-06559-t001:** The binding scores and interactions of the docked **KI** in addition to the five examined flavonoids (**1**–**5**) inside the SARS-CoV-2 Mpro pocket.

No.	Isolated Compound	S ^a^	RMSD ^b^	Interactions	Distance (Å)
**KI**	α-Ketoamide inhibitor	−8.17	1.64	Glu166/H-donor Glu166/H-acceptor Glu166/H-donor Gly143/pi-H	2.89 3.10 3.42 3.70
**1**	Taxifolin	−6.50	1.58	Arg188/H-donor Glu166/H-donor Cys145/H-donor His41/H-pi	2.85 3.16 3.60 3.44
**2**	Pectolinarigenin	−5.74	1.72	Glu166/pi-H Met165/pi-H	4.19 4.47
**3**	Tangeretin	−6.61	1.17	Glu166/pi-H Glu166/pi-H	4.09 4.19
**4**	Gardenin B	−6.48	0.74	Glu166/pi-H Glu166/pi-H	4.10 4.28
**5**	Hispidulin	−5.85	1.14	His41/H-pi Glu166/pi-H His41/pi-H	3.83 3.87 4.32

^a^ **S**: Score of a docked compound inside the docking site (kcal/mol). ^b^ **RMSD**: Root mean squared deviation between the obtained pose compared to the native one.

**Table 2 molecules-26-06559-t002:** 3D pictures showing the receptor interactions and positioning between the docked **KI** in addition to the five examined flavonoids (**1**–**5**) inside the binding site of SARS-CoV-2 Mpro.

Isolated Comp.	3D Binding	3D Positioning
**α-Ketoamide** **Inhibitor (KI)**	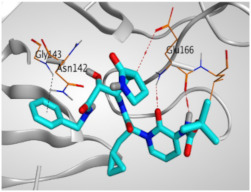	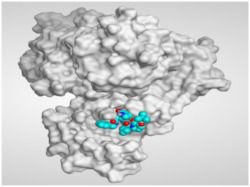
**Taxifolin** **(1)**	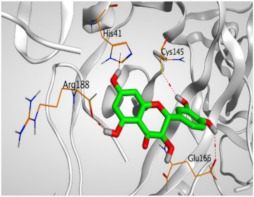	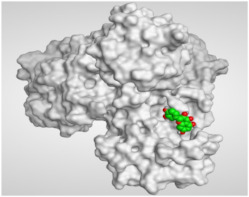
**Pectolinarigenin** **(2)**	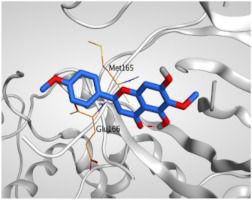	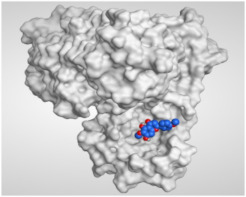
**Tangeretin** **(3)**	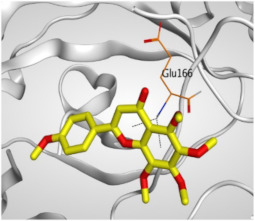	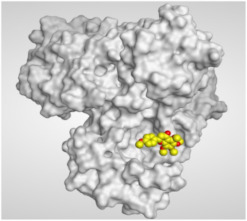
**Gardenin B** **(4)**	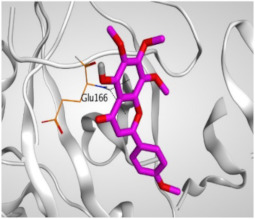	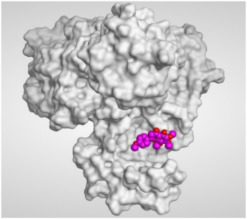
**Hispidulin** **(5)**	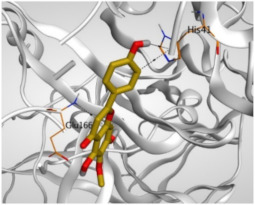	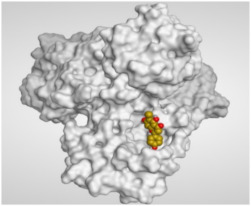

The red dash represents H-bonds and the black dash represents H-pi interactions.

## Data Availability

Not applicable.

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
