# Peer review of "Naturally Available Flavonoid Aglycones as Potential Antiviral Drug Candidates against SARS-CoV-2"

_molecules, 2021, doi:10.3390/molecules26216559_

Round 1
Reviewer 1 Report
The authors Al-Karmalawy et al., using chromatographic and spectral analysis identified five flavonoid extracts of which pectolinarigenin and tangeretin aglycones were active against SARS-CoV-2. Furthermore, they predicted it as Mpro inhibitors through molecular docking since some flavonoids inhibit this protein. It’s an interesting research, but the logic connection in the writing is hard to follow. For example, they started writing details about SARS-CoV-2, though important, it is not the main matter of the article as the title suggests. Moreover, the discussion is missing, or poor at the lines 138-143. I think a better use of references can be made to build the discussion. Thus, I do not recommend this version for publication.
Author Response
Comments and Suggestions for Authors
The authors Al-Karmalawy et al., using chromatographic and spectral analysis identified five flavonoid extracts of which pectolinarigenin and tangeretin aglycones were active against SARS-CoV-2. Furthermore, they predicted it as Mpro inhibitors through molecular docking since some flavonoids inhibit this protein. It’s an interesting research, but the logic connection in the writing is hard to follow. For example, they started writing details about SARS-CoV-2, though important, it is not the main matter of the article as the title suggests. Moreover, the discussion is missing, or poor at the lines 138-143. I think a better use of references can be made to build the discussion. Thus, I do not recommend this version for publication.
Response: The authors thank the reviewer for his efforts to improve the impact of our manuscript. Moreover, the introduction, discussion, and references were modified according to the reviewer comments to improve the quality of the manuscript. Hope the revised version will find your acceptance.
Reviewer 2 Report
Al-Karmalawy et al. reported the docking and in vitro studies of flavonoid aglycones as potential anti-SARS-CoV-2 drug candidates. The manuscript represents a nice combination of in silico and in vitro studies. However, the potency of the flavonoids evaluated in this manuscript is no better than that of the previously published flavonoids, which may lower the impact of the article. Besides, some of the methods should be more adequately described, and some of the figures should be more clearly presented.
Nevertheless, I think this work will be of value to publish in Molecules after making necessary revisions/clarification:
(1) What is the purity of the isolated compounds that were used in in vitro assays? Either NMR spectra or HPLC results should be provided.
(2) The methods used to purify the compounds should be more clearly addressed.
(3) In the method for cytotoxicity (CC50) determination section, please provide information for how vehicle control groups were set up, since DMSO used in this experiment can be cytotoxic.
(4) Line 42: The following sentence is not readable, so I suggest revising it: “at the end of 2019, Since the appearance of SARS-CoV-2; 173 vaccines are in the pre-clinical phase and 64 candidate vaccines are in clinical progress”.
(5) Figure 2 top right chart: the IC50 value, “5.412e+019” mg/mL is confusing. Please consider another way of showing this value.
(6) I would recommend providing the raw data for the docking and the in vitro assays, as part of the supplementary information.
(7) The error bars for some of the data points are missing in Figure 2. Please provide an explanation.
(8) Figure SI1: I would recommend explaining in the figure caption why different amino acid residues were color-coded differently and what do those purple glows mean (electron density?).
(9) I would suggest rewriting the sentence in line 258 as “the IC50 is the concentration of the compound required to reduce the virus-induced cytopathic effect (CPE) by 50%...”.
(10) Line 79: consider flipping the order of these two words “flavonoid” “major”.
(11) Line 114: remove parentheses.
Author Response
Comments and Suggestions for Authors
Al-Karmalawy et al. reported the docking and in vitro studies of flavonoid aglycones as potential anti-SARS-CoV-2 drug candidates. The manuscript represents a nice combination of in silico and in vitro studies. However, the potency of the flavonoids evaluated in this manuscript is no better than that of the previously published flavonoids, which may lower the impact of the article. Besides, some of the methods should be more adequately described, and some of the figures should be more clearly presented.
Nevertheless, I think this work will be of value to publish in Molecules after making necessary revisions/clarification:
(1) What is the purity of the isolated compounds that were used in in vitro assays? Either NMR spectra or HPLC results should be provided.
Response: All the isolated compounds that were used in the in vitro assays were highly pure, the NMR and HRESI–MS chromatograms were provided in the supplementary data (ESI, Figures SI1-SI10).
(2) The methods used to purify the compounds should be more clearly addressed.
Response: This part was added and explained clearly in Materials and Methods, Extraction, and isolation parts as requested.
(3) In the method for cytotoxicity (CC50) determination section, please provide information for how vehicle control groups were set up, since DMSO used in this experiment can be cytotoxic.
Response: The stock compounds were dissolved in 10%DMSO and further 10-fold diluted in 1x DMEM to a wide range of concentrations (10 ng/ml to 100 mg/ml). This means that the concentration of the DMSO in the first dilution is 1%, which is completely safe to the cells. This concentration was used for control cells. This information has been added to the “material and methods” section.
(4) Line 42: The following sentence is not readable, so I suggest revising it: “at the end of 2019, Since the appearance of SARS-CoV-2; 173 vaccines are in the pre-clinical phase and 64 candidate vaccines are in clinical progress”.
Response: The sentence has been amended as requested.
(5) Figure 2 top right chart: the IC50 value, “5.412e+019” mg/mL is confusing. Please consider another way of showing this value.
Response: The way of showing this value has been amended now in Fig 2.
(6) I would recommend providing the raw data for the docking and the in vitro assays, as part of the supplementary information.
Response: The raw data for the docking and the in vitro assays has been added to the supplementary information as requested.
(7) The error bars for some of the data points are missing in Figure 2. Please provide an explanation.
Response: The error bars for some of the data points in Figure 2 is too small to be clear on the figure. The error bars for most data points are now clearly displayed on the figure 2.
(8) Figure SI1: I would recommend explaining in the figure caption why different amino acid residues were color-coded differently and what do those purple glows mean (electron density?).
Response: The figure caption was described in detail in front of Figure SI1 as requested.
(9) I would suggest rewriting the sentence in line 258 as “the IC50 is the concentration of the compound required to reduce the virus-induced cytopathic effect (CPE) by 50%...”.
Response: The sentence was rewritten as recommended.
(10) Line 79: consider flipping the order of these two words “flavonoid” “major”.
Response: We rephrased this sentence as requested.
(11) Line 114: remove parentheses.
Response: It was removed.
Reviewer 3 Report
This work on potential RdRp inhibitors from plants has a great potential in the frame of the novel anti-COVID-19 drug discovery. However, some major points should be addressed by authors before their paper could be accepted for publication.
1-English level is generally acceptable. Nevertheless, I recommend the authors to read carefully again all section in order to improve it as I found some unclear passages.
2- Abstract should contain a reference to IC50 for KI too. This could be obtained repeating the in vitro experiments also with this inhibitor (as reference compound) or citing the literature (if the same experiments was already conducted by other scholars with KI)
3-authors should consider to express IC50 as micrograms/ml in place of mg/ml
4-More literature references on natural remedies and phytochemicals as anti-COVID-19 weapons should be cited. Add after references [3-6] at least the works with DOI: 10.1007/s10311-021-01309-5; 10.3390/sym13061041 ;10.1016/j.compbiomed.2021.104362;10.3390/md19070391; 10.1016/j.phymed.2021.153476; 10.3390/molecules26071880
5-line 72: may be 'suggest' in place of 'recommend'?
6-section 2.2: it seems that difference between the highest affinity phytochemical and KI is not so small. -6.6 vs -8.17 kcal/mol. So I am not fine with 'not largely deviated'. Use a softer way to justify the importance of your results for the phytochemicals with respect to KI. May be other powerful drugs from literature were shown to have predicted docking BE of about -6.6 kcal/mol? Please clarify
7-I understand that molecular dynamics simulation is time demanding and you based your in silico study only on docking. But consider, at least for the future to validate the results of your screening also by MDS, at least conducting it on the most active compounds. As perspective this could be also cited in the revised manuscript.
8-More details on the in vitro study should be given in section 2.3. Explain here briefly what you reported in materials and methods on the strategy of inhibition of Mpro in your tests before showing results.
9-Figure 2 is too enlarged! Fix it
10-you reported NMR characterizations for the phytochemicals. Are they given as comparison to the literature NMR data?
11- line 201: '(a-ketoamide inhibitor (KI)' should be '(a-ketoamide inhibitor (KI))'
12-line 202: 'moe file' should be 'MOE file' ? Check it
Author Response
Comments and Suggestions for Authors
This work on potential RdRp inhibitors from plants has a great potential in the frame of the novel anti-COVID-19 drug discovery. However, some major points should be addressed by authors before their paper could be accepted for publication.
1-English level is generally acceptable. Nevertheless, I recommend the authors to read carefully again all section in order to improve it as I found some unclear passages.
Response: The whole manuscript was revised carefully for any grammatical and/or typos errors and it was improved greatly.
2- Abstract should contain a reference to IC50 for KI too. This could be obtained repeating the in vitro experiments also with this inhibitor (as reference compound) or citing the literature (if the same experiments was already conducted by other scholars with KI).
Response: Thank you for your comment. KI is the co-crystallized inhibitor of the studied protein in the docking section which was used as a reference standard. However, we added the IC50 value in the abstract section from literature as requested.
3-authors should consider to express IC50 as micrograms/ml in place of mg/ml.
Response: The IC50 values were modified as requested.
4-More literature references on natural remedies and phytochemicals as anti-COVID-19 weapons should be cited. Add after references [3-6] at least the works with DOI: 10.1007/s10311-021-01309-5; 10.3390/sym13061041 ;10.1016/j.compbiomed.2021.104362;10.3390/md19070391; 10.1016/j.phymed.2021.153476; 10.3390/molecules26071880.
Response: The recommended references were cited as requested.
5-line 72: may be 'suggest' in place of 'recommend'?
Response: Ok, it was changed.
6-section 2.2: it seems that difference between the highest affinity phytochemical and KI is not so small. -6.6 vs -8.17 kcal/mol. So I am not fine with 'not largely deviated'. Use a softer way to justify the importance of your results for the phytochemicals with respect to KI. May be other powerful drugs from literature were shown to have predicted docking BE of about -6.6 kcal/mol? Please clarify.
Response: Ok, the sentence was modified as recommended.
7-I understand that molecular dynamics simulation is time demanding and you based your in silico study only on docking. But consider, at least for the future to validate the results of your screening also by MDS, at least conducting it on the most active compounds. As perspective this could be also cited in the revised manuscript.
Response: The authors thank the reviewer for his great understanding concerning MD simulations. We will consider it in the future work as suggested. Also, the importance of MD simulations was discussed and cited as requested.
8-More details on the in vitro study should be given in section 2.3. Explain here briefly what you reported in materials and methods on the strategy of inhibition of Mpro in your tests before showing results.
Response: More details have been added to section 2.3.
9-Figure 2 is too enlarged! Fix it
Response: Figure 2 has been modified as requested.
10-you reported NMR characterizations for the phytochemicals. Are they given as comparison to the literature NMR data?
Response: The compounds were isolated and their structures were elucidated by extensive chromatographic, chemical and spectroscopic methods (HRESI–MS, UV, and NMR) as well as Co-PC with reference samples. Also, their spectroscopic data were compared with previously reported values in literatures [47-50].
11- line 201: '(a-ketoamide inhibitor (KI)' should be '(a-ketoamide inhibitor (KI))'.
Response: It was modified as requested.
12-line 202: 'moe file' should be 'MOE file' ? Check it.
Response: It was modified as requested.
Round 2
Reviewer 1 Report
The authors Al-Karmalawy et al., have been improved the writing logic in this new version of the manuscript, but still there are some paragraphs that need to be clearer. At lines 92 to 94 wrote “The binding scores for the docked compounds were found to be in the following order: redocked α-ketoamide inhibitor (KI) ˃ tangeretin (3) ˃ taxifolin (1) ˃ gardenin B (4) ˃ hispidulin (5) ˃ pectolinarigenin (2)”. It seems it is referred to the table 1, but this table is not ordered in that way (it must be). Thus, the correct use of the sign “>” should be “<” as the negative numbers of the column “Sa” indicated.
Then, wrote at lines “… appeared to be good compared to that of the docked co-crystallized α-ketoamide inhibitor (-8.17 kcal/mol)”. It is true? I think the binding energies are not so good compared to the KI. I think the binding interaction with the Glu166 amino acid is enough to express the importance of the docking predictions.
Although the authors use the binding energies to explain and discuss in the docking results section, they don’t write any related to the use of binding energies in the methods docking section.
The α-ketoamide inhibitor was defined as “KI”; however, the authors use a lot of times “α-ketoamide inhibitor (KI)”, please keep it concise.
Author Response
Reviewer 1:
Comments and Suggestions for Authors
The authors Al-Karmalawy et al., have been improved the writing logic in this new version of the manuscript, but still there are some paragraphs that need to be clearer.
Response: The authors thank the reviewer for his positive impact concerning the manuscript.
At lines 92 to 94 wrote “The binding scores for the docked compounds were found to be in the following order: redocked α-ketoamide inhibitor (KI) ˃ tangeretin (3) ˃ taxifolin (1) ˃ gardenin B (4) ˃ hispidulin (5) ˃ pectolinarigenin (2)”. It seems it is referred to the table 1, but this table is not ordered in that way (it must be). Thus, the correct use of the sign “>” should be “<” as the negative numbers of the column “Sa” indicated.
Response: Thank you for your comment, but the binding scores are differing from the energy values which may be intentioned by the reviewer. The binding scores refer to the affinity of the tested compounds towards the binding pocket. So, as the binding score increases -by increasing the negative value as it is considered to be the reciprocal of binding energy-, the binding affinity increases as a result. Therefore, our order is wright, and the reviewer can check that in some of our previous publication (https://doi.org/10.3390/molecules26123772; https://doi.org/10.1039/D1RA05817G; https://doi.org/10.3390/pathogens10050623). Thanks in advance.
Then, wrote at lines “… appeared to be good compared to that of the docked co-crystallized α-ketoamide inhibitor (-8.17 kcal/mol)”. It is true? I think the binding energies are not so good compared to the KI. I think the binding interaction with the Glu166 amino acid is enough to express the importance of the docking predictions.
Response: Ok, this sentence was modified to (Their binding scores were found to be near to each other (from -6.61 to -5.74 kcal/mol) compared to that of the docked co-crystallized α-ketoamide inhibitor (-8.17 kcal/mol)) as requested from the reviewer.
Although the authors use the binding energies to explain and discuss in the docking results section, they don’t write any related to the use of binding energies in the methods docking section.
Response: The binding scores (NOT binding energies) were used as mentioned in the previous response to one of the reviewer’s comments. So, the binding score was mentioned in the docking methodology.
The α-ketoamide inhibitor was defined as “KI”; however, the authors use a lot of times “α-ketoamide inhibitor (KI)”, please keep it concise.
Response: The authors thank the reviewer for his comment. It was modified as requested.
Reviewer 3 Report
The manuscript can be published in the current form
Author Response
Reviewer 2:
Comments and Suggestions for Authors
The manuscript can be published in the current form
Response: The authors thank the reviewer for his efforts to improve the quality of our manuscript.